# Fast and accurate imputation of genotypes from noisy low-coverage sequencing data in bi-parental populations

**Cécile Triay**[1], **Alice Boizet**[2], **Christopher Fragoso**[3,4], **Anestis Gkanogiannis**[5], **Jean-François Rami**[2], **Mathias Lorieux**[1,5]*

**1** DIADE, IRD, Cirad, University of Montpellier, Montpellier, France, **2** AGAP, Cirad, INRAE, Montpellier SupAgro, University of Montpellier, Montpellier, France, **3** Verinomics, Inc., New Haven, CT, United States of America, **4** Department of Molecular, Cellular, and Developmental Biology, Yale University, New Haven, Connecticut, CT, United States of America, **5** Agrobiotechnology Unit, Alliance Bioversity-CIAT, International Center for Tropical Agriculture, Cali, Colombia

☯ These authors contributed equally to this work.

* mathias.lorieux@ird.fr

**Data Availability Statement:** All VCF data files are available at: - https://doi.org/10.23708/8FXUNC (real data) - https://zenodo.org/records/13381283 (simulated data).

## Abstract

### Motivation

Genotyping of bi-parental populations can be performed with low-coverage next-generation sequencing (LC-NGS). This allows the creation of highly saturated genetic maps at reasonable cost, precisely localized recombination breakpoints (i.e., the crossovers), and minimized mapping intervals for quantitative-trait locus analysis. The main issues with these low-coverage genotyping methods are (1) poor performance at heterozygous loci, (2) high percentage of missing data, (3) local errors due to erroneous mapping of sequencing reads and reference genome mistakes, and (4) global, technical errors inherent to NGS itself. Recent methods like Tassel-FSFHap or LB-Impute are excellent at addressing issues 1 and 2, but nonetheless perform poorly when issues 3 and 4 are persistent in a dataset (i.e., "noisy" data). Here, we present a new algorithm for imputation of LC-NGS data that eliminates the need of complex pre-filtering of noisy data, accurately types heterozygous chromosomal regions, precisely estimates crossover positions, corrects erroneous data, and imputes missing data. The imputation of genotypes and recombination breakpoints is based on maximum-likelihood estimation. We compare its performance with Tassel-FSFHap and LB-Impute using simulated data and two real datasets. NOISYmputer is consistently more efficient than the two other software tested and reaches average breakpoint precision of 99.9% and average recall of 99.6% on illumina simulated dataset. NOISYmputer consistently provides precise map size estimations when applied to real datasets while alternative tools may exhibit errors ranging from 3 to 1845 times the real size of the chromosomes in centimorgans. Furthermore, the algorithm is not only highly effective in terms of precision and recall but is also particularly economical in its use of RAM and computation time, being much faster than Hidden Markov Model methods.

**Funding:** - The French ANR project "LANDSREC" 385 (ANR-21-CE20-0012-03) - The French Government's France Génomique program through its International RIce 485 Genome INitiative "IRIGIN" project - The CGIAR Research Program "RICE" We also declare that the funders had no role in study design, data collection and analysis, decision to publish, or preparation of the manuscript.

**Competing interests:** The authors have declared that no competing interests exist.

## Availability

NOISYmputer and its source code are available as a multiplatform (Linux, macOS, Windows) Java executable at the URL https://gitlab.cirad.fr/noisymputer/noisymputerstandalone/-/tree/1.0.0-RELEASE?reftype=tags.

## Introduction

In genetic studies, bi-parental genetic populations can be created from inbred parental lines using various crossing systems, e.g., $F_2$ intercross issued from $F_1$ self-pollination ($F_2$) and recombinant inbred lines by single seed descent (SSD). These populations are used to create recombination maps and, if phenotypes are available, to find gene or quantitative-trait locus (QTL) genomic positions.

To do so, each individual of the population under study has to be characterized for its genomic content—or "genotyped" at many loci. This can be done using different molecular biology techniques, including various types of molecular markers. The gold standard for genetic variant discovery is obtained by different next-generation sequencing (NGS) techniques like restriction site-associated DNA sequencing (RADseq) [1], genotyping by sequencing (GBS) [2], and whole-genome sequencing (WGS) [3]. These techniques provide very large numbers of markers and therefore facilitate the construction of highly saturated genetic maps. This provides accurate locations of recombination breakpoints in each individual, which is important for a number of applications, e.g., studies of local recombination rate, genetic maps comparison, or QTL detection. Though NGS is less and less expensive to implement, sequencing a large number of samples can still be costly, and is commonly applied via reduced representation (RRS-NGS) or low-coverage (LC-NGS) strategies to reduce genotyping costs.

Reducing sequencing costs through minimized per-sample coverage has an important experimental downside: LC-NGS mechanically introduces a series of issues, the main ones being:

- **Issue 1**: *Low power to detect heterozygosity under low coverage*: For example, if only one sequencing read is generated at a locus, only one of the two alleles is revealed. As each additional read has a 0.5 probability of detecting the second allele, even 3 reads have only 0.75 probability of detecting a heterozygous call. Spread over thousands of sites, extensive inaccuracy in heterozygous regions becomes highly problematic.

- **Issue 2**: *Extensive genotype missingness*: The sparse distribution of reads at low coverage (3X coverage, for example, only implies an *average* of 3 reads per site) results in a complete lack of reads at some variant loci. Even in plants, which contain more genetic variation than humans, there are 6–22 SNPs per 1 Kb, resulting in abundant opportunity for non-reference variant missingness under low coverage [4].
And these issues are further compounded by those common to all types of NGS:

- **Issue 3**: *Errors due to erroneous mapping of sequencing reads*: NGS technologies are based on short reads (e.g., 150 base pair, paired-end Illumina technology). Due to the combinatorial limitation of the sequence contained in short reads, multiple mapping locations may be identified, especially in plant genomes which exhibit much more repetitive content than human genomes. Additionally, in plants, such as rice, structural variation specific to subpopulations may be completely missing in single reference genomes. These assembly errors,

omissions, and challenges posed by repetitious regions are sources of erroneous variants. Moreover, outright assembly errors may cause consistent, yet locally encountered genotyping errors.

- **Issue 4**: *Technical errors inherent to NGS methodology*: Sequencing errors may be globally introduced at a variety of stages in the NGS pipeline, from errors incurred in PCR-dependent library construction to NGS sequencing itself. The initial GBS protocol is known to generate libraries contaminated by chimeric inserts [5]. Although rare, these errors may become problematic at low coverage, as additional reads refuting an erroneous call may not be available at a given locus.

Common imputation algorithms implemented in computer programs like Beagle [6, 7] or Impute2 [8], although very accurate in diversity panels, are not well adapted to the bi-parental context since they rely on large databases to infer haplotypes. Efficient methods have been recently developed to impute genotypic data derived from LC-NGS assays in bi-parental populations. For instance, Tassel-FSFHap (thereafter simply FSFHap) [9] and LB-Impute [10] can all address issues 1 and 2 accurately. Yet, these methods can produce inaccurate results when the errors mentioned in issues 3 and 4—thereafter called "noisy data"—are too frequent. Thus, these methods might require additional bioinformatic steps to filter out low-quality markers before and after imputation. Even then, troublesome markers might not be detected easily and could alter dramatically the quality of the imputation and the final genetic map.

In this work, we present NOISYmputer, a maximum likelihood estimation algorithm for imputation of LC-NGS data that eliminates the need of complex pre-filtering of noisy data, accurately finds heterozygous chromosomal regions, corrects erroneous data, imputes missing data and precisely locates the recombination breakpoints (i.e., the meiotic crossovers). We test its accuracy using simulated data and we compare its performance with FSFHap, LB-Impute using three datasets: (1) a rice $F_2$ population sequenced by WGS, (2) a maize $F_2$ population sequenced by GBS and (3) 84 simulated $F_2$ populations with controlled depth, error rate and marker density. The algorithm is implemented in NOISYmputer, a multiplatform Java command line program (see "Availability" section).

## Design and implementation

### Imputation method

In this section we describe the main imputation algorithm, which is applied separately to each chromosome. The imputation can be preceded or followed by different filtering options in NOISYmputer (details in next section) that can be applied to reduce or eliminate the noise in the data (Fig 1).

By imputation, we mean here guessing, confirming or correcting the genotype at a SNP site in a sample. LC-NGS generates poor information in heterozygous regions (see explanation on the confounding effect in SNPs with one or few reads—issue 1 of the Introduction section). Conversely, homozygous regions are much less prone to these confounding effects. Yet, missing data (issue 2), noisiness (issue 3) and sequencing errors (issue 4) can lower the power to identify homozygous diplotypes (i.e., the combination of two gametic haplotypes). The general idea of the algorithm is, as in Hidden Markov Model (HMM), to use information of various SNPs around the imputed SNP, leaving the regions surrounding the recombination breakpoints unimputed lying between the two diplotypes. The locations of the recombination breakpoints are then inferred. Furthermore, instead of modeling error rates, we take an iterative approach to estimate them (Fig 1).

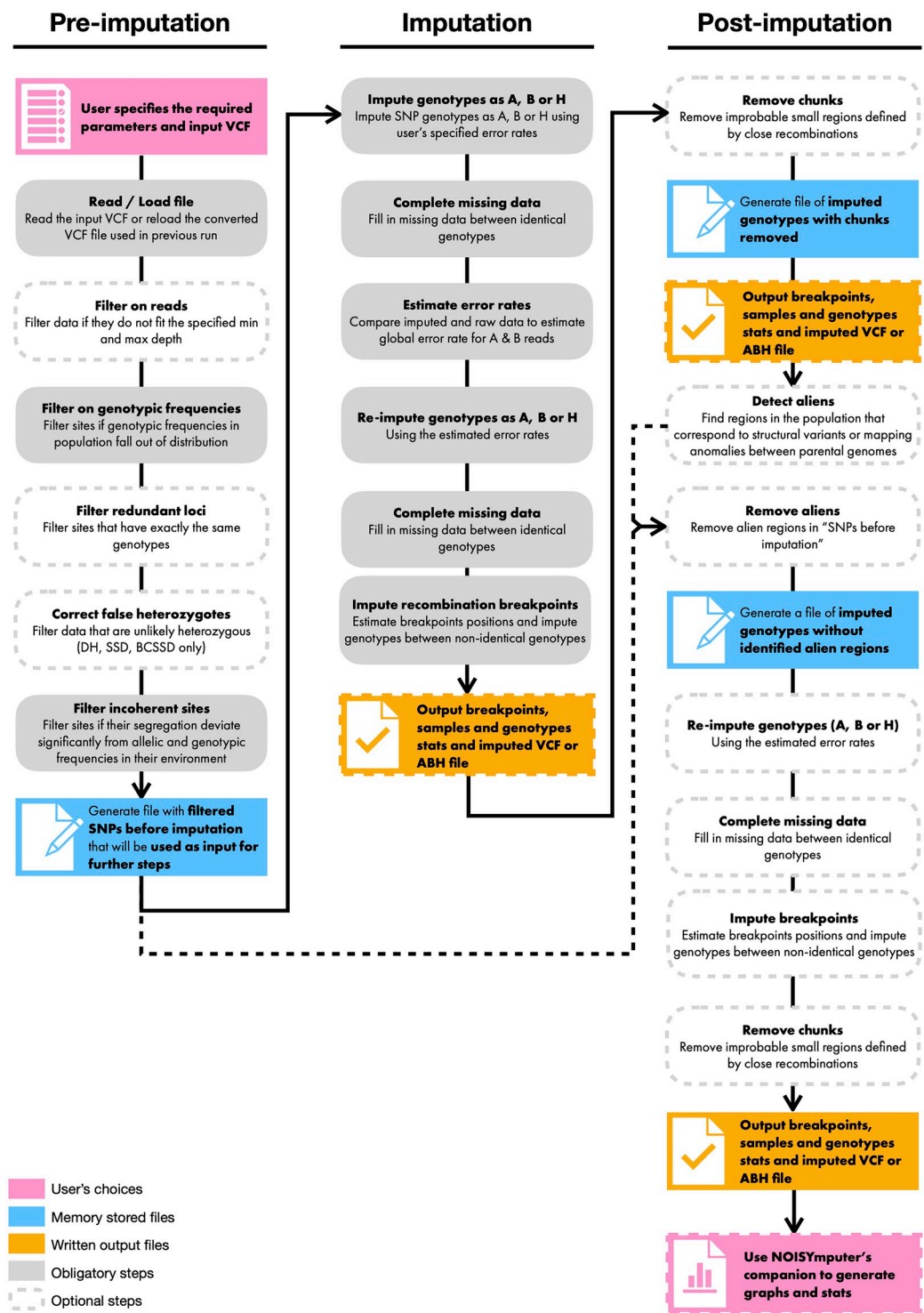

**Fig 1. NOISYmputer's workflow.** It is composed of three major phases: pre-imputation, imputation and post-imputation. Some steps are optional (dashed borders) while others are required for the algorithm to complete.

**Imputation—Step 1: Genotype calling.** Let's consider a chromosome of an $F_2$ individual with one single recombination breakpoint that separates a homozygous diplotype (AA; BB) from a heterozygous diplotype (AB, or BA, equivalent thereafter). Let's also consider a set of SNPs evenly dispersed on the physical genome, say, every 500 base pairs (bp). In the AA diplotype, and far from the breakpoint location, all SNPs should be genotyped as AA, except from the different kinds of errors cited above. To determine the genotype of a particular SNP, and due to these errors, one must consider not only its score in the VCF, but also its immediate "environment", that is, the SNPs that are located just before and just after it along the chromosome. Those surrounding SNPs help identify a potential error in the SNP scoring. Different approaches can be taken to look at the SNP environment. In segregating populations, the vast majority of the genome is exempt from crossing overs. Indeed, when implementing a sliding window method like described hereby, the expected proportion of the genome with no recombination in the window is $P_{noXO} \simeq 1 - \left(\frac{1}{100N}\right)D(8m - 2)$, where $m$ is the number of SNPs in the sliding window, $N$ is the total number of SNPs, and $D$ is the expected genome size in centimorgans (cM) (S2 File). Hence, in almost the entire genome except the breakpoint regions there are only two or three possible diplotypes, depending on the population type. Thus, instead of calculating all the likelihoods of possible paths (like in Hidden Markov Model methods), the problem is reduced to calculate the likelihoods of the data for the three possible diplotypes. Furthermore, there is no need to include transition (i.e., recombination) probabilities. The main advantage of this approach is its computation time, which increases linearly according to the diplotype size, in $O(T)$, since the log-likelihood of a diplotype is simply the sum of individual values for each site, while the time complexity is $O(T \times S^2)$ for the Viterbi algorithm applied to resolve fully connected Hidden Markov Model processes, with $T$ being the length of the sequence of observations and $S$ being the number of hidden states. We now describe the algorithm with the example of an $F_2$ population.

In practice, one defines starting values for error rates for reads A ($e_A$) and B ($e_B$), being respectively the probabilities of observing a B read ($O_B$) whereas the genotype is truly AA and observing an A read ($O_A$) whereas the genotype is truly a BB

$$e_A = p(O_B|AA) \qquad e_B = p(O_A|BB)$$

We allow different error rates for A and B reads since the A and B parents are generally not equally (i.e., genetically) distant from the reference genome. For example, one could set $e_A = 0.005$ and $e_B = 0.003$ if Parent B is closer genetically to the Reference genome than Parent A is. Those values will be automatically refined after one or several rounds of imputation.

Thus, at homozygous sites, the probability of observing an A read if the true genotype is AA is

$$p(O_A|AA) = 1 - e_A$$

and the probability of observing a B read if the true genotype is BB is

$$p(O_B|BB) = 1 - e_B$$

At heterozygous (AB) sites, and assuming that the A and B reads have the same chance to occur, the probabilities of observing A and B reads are

$$p(O_A|AB) = \frac{1}{2}p(O_A|AA) + \frac{1}{2}p(O_A|BB) = \frac{1}{2}(1 - e_A) + \frac{1}{2}e_B$$

$$p(O_B|AB) = \frac{1}{2}p(O_B|BB) + \frac{1}{2}p(O_B|AA) = \frac{1}{2}(1 - e_B) + \frac{1}{2}e_A$$

Let's consider a chromosome with $n$ SNPs. For each site $SNP_j$ of the chromosome, we define a symmetrical window ($W_j$) containing the $SNP_j$ at its center, $m$ SNPs before it in the sequence and $m$ SNPs after it (with read count $>0$). SNPs that are located in chromosome ends are omitted, since it is not possible to define symmetrical windows around them. This case is discussed later on.

For each site $SNP_i$ of the $W_j$ window three situations are possible: i) the genotype $G_i$ of the $SNP_i$ is AA (homozygous for parent A allele), ii) the genotype $G_i$ is BB (homozygous for parent B allele) or iii) the genotype $G_i$ is AB (heterozygous).

We then estimate the likelihood of observing a given combination of A reads ($n_{Ai}$) and B reads ($n_{Bi}$) at site $SNP_i$, given that the total number of reads ($n_i$) at this site is equal to $n_{Ai} + n_{Bi}$. To do so, we use the binomial distribution, with sample size at $SNP_i$ equal to $n_i$, the number of successes equal to $n_{Ai}$, and thus of fails equal to $n_{Bi}$. Knowing already the probability of observing A reads under the three possible genotypes (AA, BB and AB) we obtain the following:

$$
\begin{aligned}
P[n_{Ai}|p(O_A|AA)] &= \binom{n_i}{n_{Ai}} p(O_A|AA)^{n_{Ai}}(1 - p(O_A|AA))^{n_i - n_{Ai}} \\
&= \binom{n_i}{n_{Ai}} p(O_A|AA)^{n_{Ai}} p(O_B|AA)^{n_{Bi}}
\end{aligned}
$$

$$
\begin{aligned}
P[n_{Ai}|p(O_A|BB)] &= \binom{n_i}{n_{Ai}} p(O_A|BB)^{n_{Ai}}(1 - p(O_A|BB))^{n_i - n_{Ai}} \\
&= \binom{n_i}{n_{Ai}} p(O_A|BB)^{n_{Ai}} p(O_B|BB)^{n_{Bi}}
\end{aligned}
$$

$$
\begin{aligned}
P[n_{Ai}|p(O_A|AB)] &= \binom{n_i}{n_{Ai}} p(O_A|AB)^{n_{Ai}}(1 - p(O_A|AB))^{n_i - n_{Ai}} \\
&= \binom{n_i}{n_{Ai}} p(O_A|AB)^{n_{Ai}} p(O_B|AB)^{n_{Bi}}
\end{aligned}
$$

Since the binomial factor is the same for the three possible genotypes, it can be omitted in the calculations. Then, individual relative probabilities that the genotype $G_i$ of the $SNP_i$ is AA, BB or AB are defined as:

$$
p(G_i = X) = \frac{P[n_{Ai}|p(O_A|X)]}{\sum_X P[n_{Ai}|p(O_A|X)]}, \quad \text{with } X = AA, BB, AB
$$

The probabilities for the window's diplotype around the $SNP_j$ to be AA, BB or AB are obtained by multiplying the individual probabilities of all the SNPs in the window. As multiplication of probabilities can result in very small numbers, we add their logarithms instead to avoid reaching the precision limit of the computer:

$$
\rho_X = \sum_{i=SNP_{j-m}}^{SNP_{j+m}} \log[p(G_i = X)], \quad \text{with } X = AA, BB, AB
$$

Finally, the relative probabilities for the window's $W_j$ around the $SNP_j$ to be AA, BB or AB are

defined as:

$$P(W_j = AA) = \frac{\exp(\rho_{AA})}{\exp(\rho_{AA}) + \exp(\rho_{BB}) + \exp(\rho_{AB})}$$

$$P(W_j = BB) = \frac{\exp(\rho_{BB})}{\exp(\rho_{AA}) + \exp(\rho_{BB}) + \exp(\rho_{AB})}$$

$$P(W_j = AB) = \frac{\exp(\rho_{AB})}{\exp(\rho_{AA}) + \exp(\rho_{BB}) + \exp(\rho_{AB})}$$

A genotype is assigned to the SNP $j$ if the relative probability of its surrounding window is superior to a given threshold $\alpha$. To guarantee that no SNP is falsely genotyped, the threshold is set to a very stringent value (0.999 by default). SNPs with $P(W_j) < \alpha$ for all genotypes are assigned a missing data value. We repeat the process for each $SNP_j$ of the chromosome. For chromosome ends, the procedure is similar except that the half-window on the end side is smaller due to the lack of sites available to the left or right of $SNP_j$. This leaves two types of chromosomal regions unimputed and filled with missing data: 1) regions between imputed chromosome segments with identical diplotypes and for which none of the criteria are matched to assign a genotype, and 2) regions near recombination breakpoints.

**Imputation—Step 2: Gap filling and error rate estimation.** Step 2 consists in i) filling the unimputed regions with the surrounding genotype, with the condition that they are surrounded (left and right) by identical imputed genotypes, then ii) re-estimating error rates. The filling procedure assumes that a double recombination event is very unlikely. The maximum region size that is allowed for data filling can be calculated using the local recombination rate, which is calculated from the data of the entire $F_2$ population, imputed from Step 1. So, regions larger than the maximum size are left unimputed. It is desirable to use an interference model to estimate the distances (in cM), for instance the one implemented in the Kosambi mapping function [11]. The method employed in NOISYmputer to estimate recombination fractions in $F_2$ populations is the standard Expectation-Maximization algorithm [12].

Let's take the example of two SNPs A and C that define the bounds of such a region. They are separated by the genetic distance $d$ (cM). The maximum probability of a double crossover can be calculated as follows. We first search for the SNP B that is the closest to the middle point between A and C (in cM). Then, we calculate the recombination fractions $r_{AB}$ and $r_{BC}$ from $d_{AB}$ and $d_{BC}$ using the inverse of the Kosambi mapping function

$$r = \frac{1}{2} \tanh\left(\frac{2d}{100}\right)$$

Note that $r \approx \frac{d}{100}$ when $d < 15$ cM. In the case of highly saturated maps, this formula can be used in most intervals. Then the maximum probability of the missing data to be different to the surrounding genotype is

$$r_{ABC} = r_{AB}r_{BC} + r_{AB}{}^2 r_{BC}{}^2 \approx r_{AB}r_{BC} \quad \text{if SNPs A and C are homozygous}$$

$$r_{ABC} = r_{AB}r_{BC} + r_{AB}{}^2 r_{BC}{}^2 \approx 2r_{AB}r_{BC} \quad \text{if SNPs A and C are heterozygous}$$

The regions for which $r_{ABC} \leq \alpha$ are filled with the surrounding genotype; $\alpha$ is set to 0.001 by default. This step leaves the breakpoint regions unimputed.

We can then estimate new values for $e_A$ and $e_B$ by comparing the observed data with the newly imputed regions. This is done by simply counting the proportion of A reads in BB-imputed segments, and the proportion of B reads in AA-imputed segments.

**Imputation—Step 3: Locating recombination breakpoints.** Step 3 consists in imputing the SNP genotypes in the regions near the recombination breakpoints—i.e., between diplotypes of different states. The general idea is to determine an interval of high probability of presence (loose support interval) of the breakpoint, then to calculate the likelihood of the data under the hypothesis of a recombined segment. This procedure allows determining with high confidence a loose support interval where the recombination breakpoint is located. Here we take the example of a segment BB to the left of the breakpoint and a segment AB to the right. Since we already know from Step 1 which are the two genotypes at the left and the right of the breakpoint, we only need to consider the only two possible diplotypes, BB and AB. This saves one degree of freedom. If $k$ defines the closest SNP position to the point where $p(W_j = BB) = p(W_j = AB)$ in Step 1, we take $k − 2m$ and $k + 2m$ as starting points to guarantee that the breakpoint is covered by the interval. Then, for each $SNP_j$ of the scanned area, we recalculate $p(W_j = BB)$ and $p(W_j = AB)$, but this time in asymmetric windows of size $m$, that is, for BB, we define a window from $SNP_j$ to $SNP_{j−m}$ and for AB a window from $SNP_{j+m}$ to $SNP_j$. And then, following calculations similar to Step 1 but omitting the probabilities for the AA genotype:

$$p(W_j = BB) = \frac{\exp(\rho_{BB})}{\exp(\rho_{BB}) + \exp(\rho_{AB})} \quad \text{in the B window}$$

$$p(W_j = AB) = \frac{\exp(\rho_{AB})}{\exp(\rho_{BB}) + \exp(\rho_{AB})} \quad \text{in the H window}$$

Starting from $k − 2m$, and progressing to the right, we look for the first site $SNP_j$ for which:

$$P_{SI} = (1 − p(W_j = BB))((1 − p(W_j = AB)) > \alpha_{SI}$$

with $SI = 0.05$ by default.

The explanation for the calculation of PSI is provided in S3 File.

The breakpoint loose support interval is defined between the first position from the left ($k_L$) and from right ($k_R$) where $P_{SI} > SI$. The breakpoint support interval and position are then estimated within the loose support interval. To do so, for each $SNP_j$ in the breakpoint interval $k_L$ to $k_R$, a probability $P_{bkp}$ that the diplotype's window contains a breakpoint in its middle is estimated. We define a left window for $p_{bkp}(W_j = BB)$ that includes the $SNP_j$ and goes to the left until the window's data count reaches $m/2$ SNPs with at least one read (the left boundary of this window is called $m_L$) and a right window for $p_{bkp}(W_j = AB)$ that starts at $SNP_{j+1}$ and goes to the right until the window's data count reaches $m/2$ SNPs with at least one read (the right boundary of this window is called $m_R$). Values of $m_L$ and $m_R$ are recalculated for each $SNP_j$.

The log-probabilities for the left and right segments are:

$$\rho_{bkp}(W_j = BB) = \sum_{i=m_L}^{j} \log[p(G_i = BB)]$$

$$\rho_{bkp}(W_j = AB) = \sum_{i=j+1}^{m_R} \log[p(G_i = AB)]$$

Then, the probability that the $SNP_j$ and $SNP_{j+1}$ are surrounding the breakpoint is:

$$p_{bkp}(BK_j) = \exp(\rho_{bkp}(W_j = BB) + \rho_{bkp}(W_j = AB)$$

And after normalization:

$$P_{bkp}(BK_j) = \frac{p_{bkp}(BK_j)}{max(p_{bkp}(BK_z) : z = k_L, \ldots, k_R)}$$

The breakpoint is estimated in the middle of the interval defined by the SNP having the maximal $P_{bkp}(BK_j)$ and the next SNP to its right.

Finally, the unimputed genotypes in the breakpoint area are completed in assigning the BB genotype to the SNPs to the left of the SNP with the max $P_{bkp}(BKj)$ (included) and AB to the right. Imputation of breakpoint positions for the other types of homozygous-heterozygous transitions (AB→BB, AA→AB, AB→AA) are easily derived from the example beforehand. The support interval for the breakpoint around its most likely position can be defined in searching for the SNPs (left and right starting from the SNP with the maximum $P_{bkp}(BK_j)$ for which $-\log_{10}(P_{bkp}(BK_j) \geq \alpha_{drop}$, where $\alpha_{drop}$ is the dropping value of $P_{bkp}$. $\alpha_{drop}$ is set to 1 by default, corresponding to ten-fold decrease of $P_{bkp}$ compared with $P_{bkp}(BK_j)$.

## Filtering options—Before imputation

**Genotypic frequencies, heterozygosity, missing data.** The program can filter out SNPs for parental genotypes, and progeny heterozygosity, percentage of missing data and parental genotypic frequencies. Min and max filtering values can be manually entered (though usually not recommended), or the program can calculate them from the genotype matrix imported from the VCF. In this case, genotypic frequencies are calculated for each SNP, and the filter values are derived from the extreme percentiles of the frequency distribution. Correction factors can be applied to the percentiles, to avoid too small or too large values.

**Read counts.** SNPs with too few or too many reads can be eliminated. This can be useful to, for instance, remove SNPs in duplicated regions. By default, when reading the input VCF file, if the depth of a site for a sample is at least 10 times superior to the average depth of this same sample (across all sites), the genotype at this site is set to a missing data. Also by default, no filter is applied on the minimum number of reads required to consider a site in a sample.

**Incoherent SNPs.** In sequence-based genetic mapping, it is common to observe SNPs that do not segregate the same way as their immediate environment, indicating a probable mapping error due to, for instance, structural variation between the reference genome and the population parents, or between the parents, or both. As segregation distortion is a frequent phenomenon in many organisms, the Mendelian expected frequencies cannot be used to analyze the SNP segregation. Instead, the procedure defines a window of $n$ SNPs around each tested locus. By default, $n$ = 1% the number of SNPs in the largest chromosome. For each window/SNP couple, it calculates the genotypes AA, BB and AB frequencies and the reads A and B frequencies across the population from the genotypes called in the VCF and compares the SNP with the window segregation of genotypes and reads using a chi-square test, where expected counts are the observed frequencies in the window multiplied by the population size. It then filters out SNPs for which the chi-square statistic exceeds a defined threshold for genotypes or reads frequencies.

### Filtering options—After imputation

**Incoherent chromosome segments (single individual).**   Even after imputation and the different filtering operations, some few, improbable chromosome short diplotypes can still remain in the imputed matrix—we call them "small chunks". The procedure identifies each small chunk composed of identical alleles, embedded in a homogeneous genomic environment that has a different allele. The method resembles the one used in Imputation—Step2.

Consider two SNPs A and C that define the bounds of a region imputed as H and surrounded by regions imputed as A or B. Search for the SNP B that is the closest to the middle point between A and C (in cM). Also search for an SNP D before the SNP A so that $d_{DA} \approx d_{AB}$, and an SNP E *after* the SNP C so that $d_{CE} \approx d_{BC}$.

Then, calculate the recombination fractions $r_{DB}$ and $r_{BE}$ from $d_{DB}$ and $d_{BE}$ using the inverse of the Kosambi mapping function. Then the maximum probability of the "chunk" to be different to the surrounding genotype is

$$r_{ABC} = r_{DB} r_{BE}$$

The chunks for which $r_{ABC} \leq \alpha$ are restored with the surrounding genotype; is set to 0.001 by default.

**Incoherent chromosome segments (cross-population).**   Entire chromosome segments can be misplaced due to different kinds of genomic structural variation such as translocation, or duplication in one of the two parents that is not present in the reference genome. Such segments are called "aliens" in the program. If their size is too large, the chi-square procedure that filters out the incoherent SNPs may fail to identify them since it is run *before* the imputation. Alien segments are easily detected, as they produce severe map expansion. The procedure searches for SNPs that mark rapid changes in the slope of the cumulated centimorgans of the genetic map calculated from the imputed matrix. If a SNP marker is detected, the procedure then searches for the next SNP that is closely linked (by default $r < 0.01$) to the SNP located just before the slope change. It then eliminates all the SNPs that are in-between.

## Running the program

### Algorithm implementation

The program is implemented in Java, as a Spring Boot (v2.6.7) project. Spring Boot is an open-source Java framework used to create standalone java applications. The executable .jar has been built with JDK 8 using Maven (v3.9.6), an open-source build tool.

Paths to datafiles and working folders paths, as well as parameters for imputation and filtering can be entered in a config file or directly in the command line. A "NOISYmputerResults" folder is automatically created, where the program writes all the output files.

### Data specifications

In this current version, NOISYmputer is built and extensively tested to perform on $F_2$ intercross data, that is, the progeny from $F_1$ self-fertilization ($F_2$). NOISYmputer can also be used on recombinant inbred lines by single seed descent from the $F_2$ (SSD).

Input data for NOISYmputer are standard Variant Call Format (VCF) files, with chromosome coordinates. Genotypes (GT field) and allele depths (AD field) must be present in the VCFs. The data can be low coverage, that is, the sum of all sequences produced per sample is equivalent to 1–3 times (1–3 X) the size of the reference genome used. Ideally, the VCF should contain only bi-allelic single-nucleotide polymorphisms (SNPs), however NOISYmputer automatically filters out the other types of sites. Small indels are not handled. Parental lines need to

be included in the VCF file with the prefix "Parent" in their name. Compressed ".gz" VCFs are accepted.

## Results

NOISYmputer, FSFHap and LB-Impute were run on the IFB Core cluster (specs. available at https://ifb-elixirfr.gitlab.io/cluster/doc/cluster-desc/ and in details in S1 File) with one allocated node per job and 32GB to 64GB of RAM to make sure that the tested programs are fully efficient.

Details on parameters used for the three imputation methods are provided in S1 File.

### Using simulations for calibration

To test NOISYmputer's efficiency in precision and recall of breakpoints estimation, we used simulated $F_2$ datasets generated using PopSimul (https://forge.ird.fr/diade/recombination_landscape/popsimul). A set of 84 VCFs with n = 300 samples and varying values of marker density, mean depth and error rate were generated for a final expected map size of 180 cM (corresponding to an average of 3.6 breakpoints per sample) to mimic the chromosome 1 of rice (available at https://zenodo.org/records/13381283). Using five different imputation window sizes, we compared the outputs of NOISYmputer to the known positions of breakpoints in the simulated data. In total, a set of 420 combinations were analyzed. All combinations and tested parameters are listed in Table 1. The results of these analyses confirmed that NOISYmputer efficiently detects the recombination breakpoints and precisely estimates their positions.

**Breakpoint precision-recall.** We assessed NOISYmputer's ability to correctly detect all breakpoints within samples by comparing positions of breakpoints found by NOISYmputer to those of simulated datasets. We considered a breakpoint correct when the simulated breakpoint position falls within NOISYmputer's loose support interval, along with the correct transition type.

Across all 420 VCFs, representing an average of 455,000 breakpoints, NOISYmputer demonstrated robust recall score, correctly finding 99.5% of simulated breakpoints (median at 99.6%). NOISYmputer also displayed high precision as, on average, 98.9% of breakpoints identified correspond to actual breakpoints (with a median at 100%). Thus, NOISYmputer presents an overall excellent precision and recall in detecting breakpoints.

To better understand the impact of each parameter and their interaction on NOISYmputer performance, we performed a principal component analysis (PCA) on parameters and performance indicators. Precision was primarily influenced by error rates, but was also affected by the imputation window size when excessively large. Conversely, smaller window sizes enhanced recall. Also, higher marker density correlated with improved recall, as lower

**Table 1. Parameter values used in PopSimul to generate simulated $F_2$ VCFs: Marker density, mean depth and error rate.** All possible combinations of these parameters were tested and imputed using a range of imputation windows in NOISYmputer.

| Parameters | Marker density (in number of markers along the chromosome) | Mean depth (in X) | Error rate | NOISYmputer impute half window size |
|---|---|---|---|---|
| Tested values | 220,000 | 0.5 | 0.05 | 15 |
| | 180,000 | 1 | 0.01 | 20 |
| | 100,000 | 1.5 | 0.005 | 30 |
| | 66,000 | 2 | | 50 |
| | | 2.5 | | 100 |
| | | 3 | | |
| | | 4 | | |

densities limit NOISYmputer's ability to identify breakpoints in regions with high recombination rates.

Some specific combinations decreased NOISYmputer precision and/or recall but overall the lower performances were still good. For instance, the lowest precision was of 72.3% (with error rates at 0.05 and smaller imputation window size of 15), and the lowest recall was of 96.9% (with larger imputation window size of 100). This is expected as small windows with high levels of noise are prone to false positive breakpoints. Nevertheless, when setting the window size to a larger value—30 –, NOISYmputer achieves excellent results even with error rates as high as 5%. As such, the minimum precision is 96% and the recall is 99%, regardless of marker density or depth, even with reads with 5% error rate (S1 Table). On the other hand, large windows (especially if coupled with low depth or marker density) may miss double recombination events, leading to false negatives (Fig 2A and 2C). In more realistic conditions, error rates as high as 0.05 are not typically observed in Illumina sequencing and alignments. When removing runs with the 0.05 error rate, the average breakpoint precision reached 99.9%, with a median of 100%. Similarly, the average recall was 99.6%, with a median of 99.5% (Fig 2B, S1 Fig).

The data in the VCF files, such as sequencing depth or marker density or species model, depend on the model species or sequencing type and are generally not under the user's control. We thus looked for the imputation window size producing the best results for both breakpoint

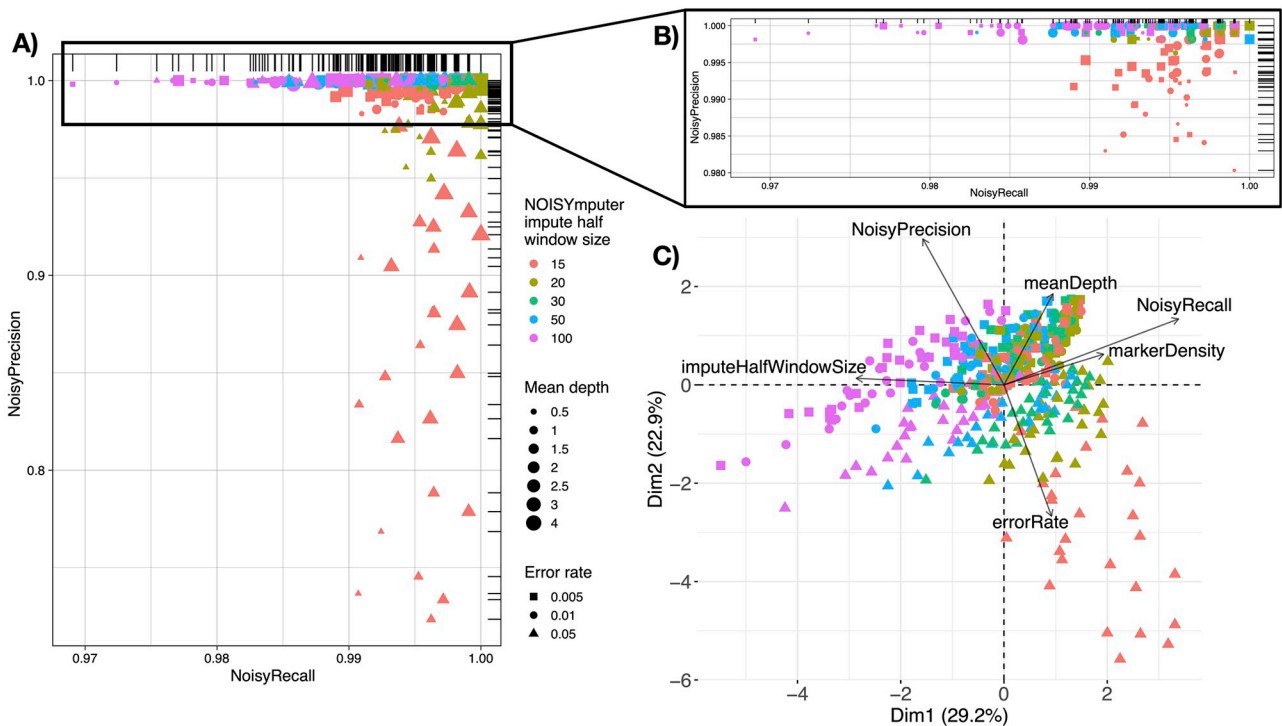

**Fig 2. Most impacting parameters and data characteristics on NOISYmputer results based on 420 outputs of simulated VCFs of F₂ populations.**
**A)** Representation of NOISYmputer's precision (proportion of NOISYmputer-identified breakpoints being actual breakpoints from simulated data) in function of NOISYmputer's recall (proportion of simulated breakpoints correctly identified by NOISYmputer). NOISYmputer shows excellent recall and precision with at least 72.3% and 96.9% respectively. **B)** Zoom on the upper part of the A plot of precision and recall, ignoring the error rate of 0.05. **C)** PCA Biplot of NOISYmputer showing VCFs characteristics and imputation window size influencing precision and recall with simulated VCFs. The lowest recalls are observed when high error rates are coupled with a small imputation half-window size in NOISYmputer. The lowest precisions correspond to VCFs imputed with a large imputation half-window size in NOISYmputer and can be accentuated by very low depth (≤ 1X) and/or low marker density (< 66, 000 sites /44 Mb).

precision and recall with the VCF that mimicked best the real $F_2$ rice data we had. In both cases, the optimal results were obtained by the imputation half window size of 30. Thus, we used this value of 30 later on when exposing NOISYmputer to real datasets.

**Precision of breakpoint position.** NOISYmputer's precision of breakpoint position was estimated by computing the difference between the simulated breakpoints positions and the estimated ones by NOISYmputer. We considered the size of the support interval and its marker density to estimate discrepancy (in number of SNPs) with the actual breakpoint position.

Across all 420 VCFs, a difference of 1,427 bp on average (equivalent to a discrepancy of $\sim$ 2 SNPs) was observed. The median difference was even lower, with only 245 bp ($<$ 1 SNP discrepancy). This disparity between the median and mean is mainly due to extreme combinations, particularly low depth combined with high error rate. Notably, variance is higher in 0.5X coverage VCFs, becoming more homogeneous at 1X coverage.

Regarding the imputation window, smaller half-windows resulted in lower average differences between NOISYmputer and simulated positions but increased the median difference. Consequently, smaller windows enhanced overall precision of breakpoint position while potentially increasing the occurrence of extreme discrepancies.

**Error rate estimations.** Error rates ($e_A$ and $e_B$), are recalculated after a first iteration of imputation step 1. NOISYmputer correctly estimated the error rates in 100% of the cases, with an average difference between simulated and estimated error rates of 9.8 $10^{-7}$ (standard deviation 4.8 $10^{-5}$) (S2 Table). This reflects the accuracy of the imputation, even with starting values for error rates far from the true values.

## Confirmed efficiency on real data and comparison with other methods

We assessed the performance of NOISYmputer on two real datasets: i) a maize $F_2$ population in GBS with 91 samples, including the parents, and ii) a rice $F_2$ population with 3X coverage in whole-genome sequencing (WGS) comprising 222 samples, including the parents sequenced at $\sim$ 30X. Details of how the real dataset for rice was generated are summarized in the S1 File and the data are available at https://doi.org/10.23708/8FXUNC). The maize dataset is described in the LB-Impute publication [10].

In real data, direct estimation of imputation accuracy may be challenging due to the unknown true state at each locus. However, it is possible to assess the quality of the imputation indirectly by comparing the final genetic map to, for instance, existing high-quality maps. A correctly imputed dataset should yield a map size—in centimorgans (cM)—consistent with those derived from high-quality marker data. Conversely, datasets with a high rate of genotyping errors will exhibit map expansion, resulting in a longer genetic map due to falsely imputed recombination breakpoints.

Using map size estimates in centimorgans (cM) of chromosome 1 of these datasets, we compared the results of NOISYmputer to those of LB-Impute and FSFhap (Fig 3 and Table 2). Concerning the maize GBS dataset, LB-Impute and FSFhap strongly overestimated the map size expected from high-quality datasets (respectively 633 cM and 13,271 cM), whereas NOISYmputer's map was in range with the expected map size (203 cM). Regarding the Rice WGS dataset, while both LB-Impute and FSFhap yielded maps much larger than expected (23,436 cM and 337,750 cM respectively), NOISYmputer estimated a map size close to the expected value (208 cM). The lack of prefiltering only partially explains the poor performance of LB-Impute and FSFhap: even when providing a VCF prefiltered with NOISYmputer, they still produce extremely long maps (11,571cM and 47,319 cM respectively) (S3 Table). Also, results from FSFhap on PopSimul data produced extremely large map size estimations for high error

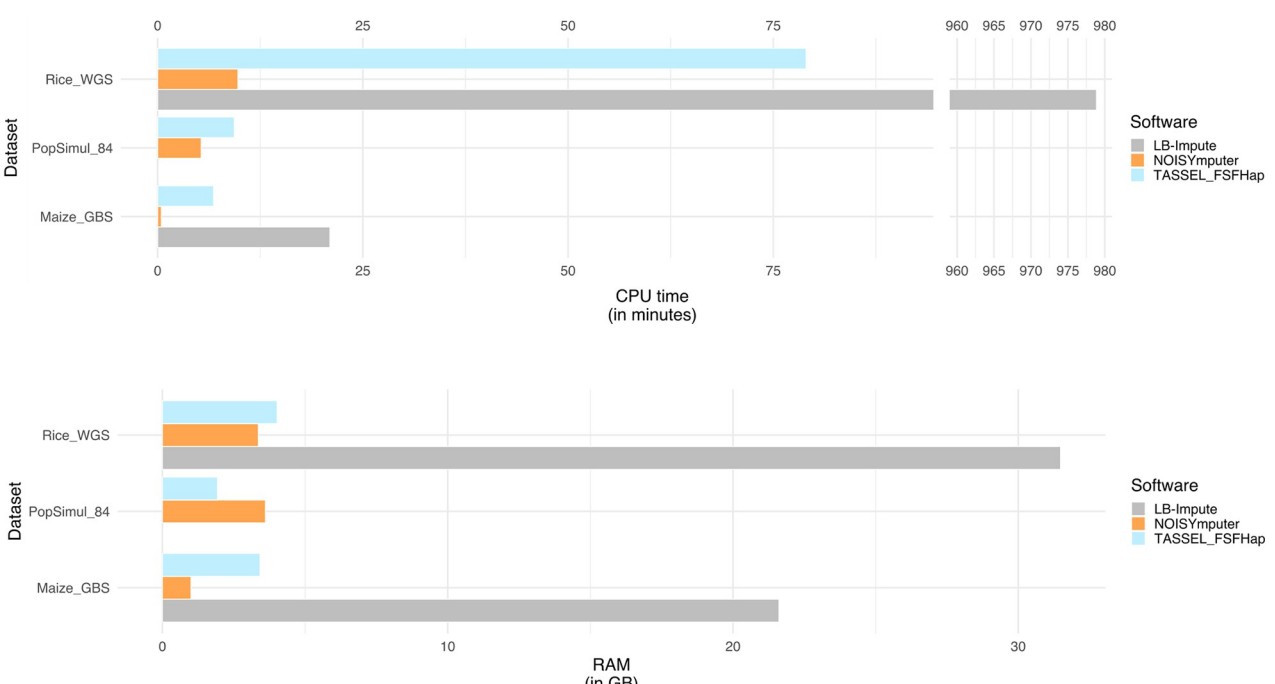

**Fig 3. Barplot of CPU time and RAM resource usage for NOISYmputer (orange), LB-Impute (gray) and FSFHap (blue) on three datasets.** Rice_WGS is an $F_2$ Rice WGS dataset with n = 222 samples including parents; PopSimul_84 values are averages across 84 VCFs generated with PopSimul, each VCF containing n = 300 samples including parents, simulated using ranges of depth, marker density and error rate to mimic different characteristics of $F_2$ VCFs; Maize_GBS is an $F_2$ Maize GBS dataset with n = 91 samples including parents (with a lower marker density than Rice_WGS). NOISYmputer is overly faster and more RAM-efficient in all conditions than FSFHap and LB-Impute, with the exception for RAM usage on simulated VCF files of PopSimul. No data is shown for PopSimul_84/LB-Impute, as LB-Impute was not benchmarked due to excessive CPU time.

rates (5%) VCF (1,397,752 cM on average), whereas NOISYmputer estimates the map size to be 180 cM compared to the 180 cM expected. Results were qualitatively similar between NOISYmputer and FSFHap for lower (1% and .5%) error rates (S1 Table).

To further estimate the performance of NOISYmputer on real datasets we also performed comparisons on precision, recall, and breakpoint position estimate in the $F_2$ Rice population

**Table 2. Comparison of estimated and expected map sizes for three different datasets using NOISYmputer, FSFHap and LB-Impute.** The 84 VCFs generated using PopSimul have varying numbers of markers (66,000, 100,000, 180,000 or 220,000), depending on the settings used to generate the VCFs. Overall, NOISYmputer is showing considerably higher accuracy in map size estimation compared to FSFHap and LB-Impute. Estimated map size is given as average in the case of the 84 simulated populations.

| Dataset | Software | Estimated map size (cM) | Expected map size (cM) | Initial number of markers in VCF |
|---|---|---|---|---|
| $F_2$ Maize GBS $n$ = 91 samples including parents | NOISYmputer | 203 | | |
| | FSFhap | 13,271 | 200 | 17,945 |
| | LB-Impute | 633 | | |
| $F_2$ Rice WGS $n$ = 222 samples including parents | NOISYmputer | 208 | | |
| | FSFhap | 337,750 | 183 | 254,095 |
| | LB-Impute | 23,436 | | |
| 84 PopSimul $F_2$ populations $n$ = 300 samples each including parents | NOISYmputer | 180 | | |
| | FSFhap | 479,399 | 180 | Different numbers of markers depending on the simulation settings |
| | LB-Impute | N/A | | |

($n$ = 222). This dataset includes 20 samples sequenced at $\sim$20X depth, and artificially sub-setted to 3X (that we call pseudo-3X). These 20 samples allow for a more robust evaluation as their breakpoints are well estimated thanks to their better depth. We processed similarly to the simulated analyses and compared breakpoint precision, recall, and precision of breakpoint position for NOISYmputer against the accurately estimated breakpoints at 20X coverage. Unfortunately, we were not able to compare NOISYmputer results to those of FSFHap and LB-impute as, even if we managed to retrieve each breakpoint position estimate, we could not easily check which were actual breakpoints and which were false positives, as TASSEL FSFHap and LB-impute do not provide support intervals for breakpoints.

Overall, NOISYmputer demonstrated excellent results with, on average, 99% precision and 97% recall. Regarding precision, on average the difference in position was of 10,219 bp, while the median was of only 415 bp. The large difference between the average and the median is due to a few breakpoints estimated far from their true position. Indeed, 80% of the breakpoints were still estimated at less than 1,669 bp from their true position. In terms of number of SNPs, the discrepancy was of 2 SNPs on average (median: 1) (S4 Table).

Overestimation of map sizes was mostly due to misinterpretation of noisy data by FSFHap and LB-Impute. These discrepancies frequently arise in regions corresponding to structural variations between parental genomes. Such variations can occur, for instance, when attempting to map onto regions found exclusively in the Parent A genome, which serves as the reference. In such cases, reads from B regions might map to the most similar A regions available resulting in false recombination events according to imputation softwares. This phenomenon is accentuated in WGS data compared to GBS data as the complete genome is sequenced and mapped, thus increasing the number of markers. Including more sites, inducing sites belonging to peculiar genomic structures, can hinder the quality of imputation if the software does not take into account the coherence of a marker with its surrounding environment in the population. Though FSFHap and LB-impute might be precise in the estimated breakpoints positions, their lack of precision in breakpoints detection leads to results, on whole genome datasets, difficult to use without the help of complex filtering steps. NOISYmputer, on the contrary, is very efficient at correcting mapping issues or divergence between parental genome structures. Moreover, the filters applied by NOISYmputer allow keeping substantial amounts of data, as shown in S5 Table. For instance, in the case of the Rice WGS dataset, nearly 150,000 SNPs are kept for chromosome 1, allowing the generation of an extremely saturated map.

## A resource-optimized software, CPU- and RAM-efficient

In our comparative analysis of the NOISYmputer with established counterparts, we conducted comprehensive benchmarks, focusing on execution time and RAM usage (Table 3 and Fig 3). To do so, we ran NOISYmputer, FSFHap and LB-Impute on simulated and real datasets. We then retrieved their CPU time, "wall clock" execution time and RAM usage using the *seff* command on the IFB computing cluster.

Concerning the $F_2$ Maize GBS dataset, NOISYmputer ran $\sim$10 and $\sim$45 times faster than FSFhap and LB-Impute, respectively. It also used less RAM ($\sim$3.4 GB), $\sim$3 times less than FSFHap and $\sim$21 times less than LB-Impute.

Regarding the $F_2$ Rice WGS dataset, NOISYmputer used slightly less RAM than FSFHap and was $\sim$13 times faster (<6 min vs. 1h19m). LB-Impute showed poor CPU and RAM efficiency as NOISYmputer used $\sim$9 times less RAM and ran $\sim$145 times faster.

Due to the excessive computation time on this single smaller dataset, LB-Impute was excluded from the remaining comparisons with the 84 PopSimul VCFs with 300 samples. It is interesting to note that FSFHap resource efficiency is better on simulated than on real datasets

**Table 3. CPU and RAM usage of NOISYmputer, FSFHap and LB-Impute for three datasets based on the output of the *seff* command on the IFB cluster.** NOISYmputer 1st and 2nd Runs are displayed as NOISYmputer shows better CPU time usage for the second run since the conversion of the raw VCF file has already been done. For LB-Impute, as imputation is processed in two steps, CPU time and execution time results are the sum of the two steps; RAM usage corresponds to the highest RAM usage of the two steps (offspring imputation). For the 84 PopSimul VCFs section, results correspond to the average of resource usage for each of the 84 PopSimul VCFs for an imputation half-window of 30 SNPs with NOISYmputer and the default window size (50) of FSFHap. All tests were conducted on the IFB Core cluster. *LB-Impute showed excessive time/RAM use so it could not be evaluated.

| Dataset | Software | CPU time (h:m:s) | Total execution time (h:m:s) | RAM (GB) |
|---|---|---|---|---|
| $F_2$ Maize GBS $n$ = 91 samples including parents | NOISYmputer—1st run | 00:00:27 | 00:00:28 | 1.00 |
| | NOISYmputer—2nd run | 00:00:07 | 00:00:09 | 1.00 |
| | FSFhap | 00:06:49 | 00:04:35 | 3.42 |
| | LB-Impute | 600:20:59 | 00:21:04 | 21.61 |
| $F_2$ Rice WGS $n$ = 222 samples including parents | NOISYmputer—1st run | 00:09:47 | 00:06:44 | 3.36 |
| | NOISYmputer—2nd run | 00:06:49 | 00:04:35 | 3.42 |
| | FSFhap | 01:19:00 | 01:19:06 | 4.02 |
| | LB-Impute | 16:18:52 | 16:19:21 | 31.48 |
| 84 PopSimul $F_2$ populations $n$ = 300 samples each including parents | NOISYmputer—1st run | 00:05:18 | 00:05:39 | 3.61 |
| | FSFhap | 00:09:20 | 00:09:24 | 1.93 |
| | LB-Impute* | N/A | N/A | N/A |

even though they have more samples. Indeed, FSFHap used on average 1.93 GB of RAM, whereas NOISYmputer was stable at 3.61 GB. NOISYmputer was still faster than FSFHap on average, with ∼ 5 min, while FSFHap ran in ∼ 9 min. This underlies the difficulty that FSFHap has to impute noisy data, partly due to structural variants and calling errors. These results underscore NOISYmputer's efficiency improvement in processing imputation tasks, especially compared to existing software for bi-parental population imputation.

## Availability and future directions

### Availability

NOISYmputer is available as a multiplatform (Linux, macOS, Windows) Java executable at the URL https://gitlab.cirad.fr/noisymputer/noisymputerstandalone/-/tree/1.0.0-RELEASE?ref_type=tags. The source code and the documentation are available at the same URL. A Quarto markdown companion (compatible with R markdown and Jupyter notebooks IDE) that allows to display graphics of statistics (e.g., genotypic frequencies on SNPs and samples) and graphical genotypes from NOISYmputer output files was developed and is also available.

NOISYmputer and its companion are distributed under the GNU Affero General Public License V3.0.

The 84 simulated VCFs that mimic the chromosome 1 of rice are available at https://zenodo.org/records/13381283.

### Future directions

**NOISYmputer's strengths.**   Although previous methods have made significant advances in addressing the challenges listed above, the noisiness of imputed datasets are still producing expanded genetic maps, excess heterozygosity, and probabilistically unlikely recombination events contained within a short physical interval. Here, we introduce an algorithm which, in a series of steps, addresses each source of error to create higher-quality datasets for improved trait mapping and genomics-assisted breeding. Our algorithm represents a step to systematically address all sources of NGS genotyping error and even errors in the reference genome,

and hopefully the corrections brought here will be integrated into future algorithm development. Indeed, key features of NOISYmputer are its pre- and post-filtering steps that other currently available software does not perform. In filtering SNPs and segments that are incoherent with their environment and with the population local recombination landscape, NOISYmputer efficiently eliminates errors of genotype calling, sequencing errors, or errors generated by structural variants. The pre-imputation and post-imputation stages of NOISYmputer, in particular, address artifacts of imputation caused by presence-absence variation misrepresented by the reference assembly and assembly errors from inaccurate or misordered contigs. These imputation artifacts, such as those caused by collapsed structural variants (incoherent sites or false heterozygosity) or misassembled "chunks", are not systematically addressed by other imputation methods, such as LB-Impute [10], and otherwise must be parsed through manual filtering of the imputed dataset.

NOISYmputer is a resource-effective software developed in Java, allowing its integration in bioinformatics pipelines. NOISYmputer is parallelizing computation at the sample level in several steps of the algorithm, which increases its speed considerably. The use of a Java standalone executable also allows to simulate parallelization in running each chromosome on a separate core of a server/cluster. Moreover, NOISYmputer employs a maximum likelihood method, instead of hidden Markov models, which considerably reduces computational complexity, compared to FSFHap [9] and LB-impute [10], while enhancing result accuracy and flexibility across diverse datasets. Indeed, NOISYmputer is less sensitive to noisy regions (due to mapping artifacts for example) as it can handle large windows without being greedy in RAM and computation time to overpass complex regions.

Notably, NOISYmputer's speed allows iterative refinement of parameter settings. For example, the size of the imputation window (in number of SNPs), like in other imputation programs (e.g., FSFhap, LB-Impute), is arbitrarily fixed by the user. The most appropriate value for m depends on several factors, including depth and SNP density. A convenient way to determine which value for m to use is to run the imputation several times with different values until reaching the expected distribution of the number of recombination breakpoints per sample across the population (if previously known). Often, saturated genetic maps generated with other types of markers are available in the literature, from which the expected distribution is easily derived. With our rice data, the imputation algorithm gave the best results with $m = 30$, so even a few runs should provide a satisfying window size.

Furthermore, NOISYmputer generates a .json file from the VCF during the initial run, that is used by the consecutive runs, eliminating the redundant tasks of converting the input VCF file, thus enhancing speed for subsequent launches on the same dataset.

Its robust performance extends to various VCF characteristics, accommodating differences in SNP quality, marker density, error rates, and sequencing depths. This is partly due to its low sensitivity to the SNP calling step used to generate the input VCF, as NOISYmputer is re-estimating the probabilities of genotypes using the allele depth at each site, along with information of the surrounding environment and of the whole population. This results in maintenance of overall excellent precision, recall and position precision on recombination breakpoints even with very low coverage datasets ($\leq$1X). However, users should exercise caution in selecting an appropriate imputation window size to mitigate the risk of false positives and negatives.

In addition to its performance benefits, NOISYmputer provides users with several comprehensive breakpoint confidence information allowing to further filter the identified breakpoints. This is a feature that is innovative and useful and not available in other software, to our knowledge. NOISYmputer also outputs statistics on genotypic/allelic frequencies, samples and genetic map among others.

**Suggestions for improvement.** NOISYmputer could benefit from several improvements. The first one is including more population types. In the next version, we will implement $F_2$ backcross, or $BC_1F_1$, the progeny of the $F_1$ hybrid crossed with one of the parents ($BC_1$); doubled haploid of $F_1$ gametes (DH); $F_2$ intercross, that is, the progeny from $F_1$ self-fertilization ($F_2$); recombinant inbred lines by single seed descent from the the $BC_1F_1$ (BCSSD); the unconventional mating design (UMD) $BC_1F_3$, derived by two generations of self-fertilization of $BC_1F_1$ individuals. For now, it has been extensively tested and optimized for $F_2$ crosses between distant parents which might be one of the hardest designs to estimate breakpoints from. We thus are confident that the algorithm can be adapted to these other types of crosses.

Breakpoint precision and recall could benefit from a more complex modeling of the likelihood. Currently, we test for the existence of a single transition within the loose support interval in imputation Step 3. Testing for one, two or even three transitions in a single interval could increase the probability of finding close double recombination events if they happened to have a higher probability in the tested region. Breakpoint position estimation, on the other hand, might be improved by using a combination of NOISYmputer's current algorithm with a hidden Markov model occurring in the Step 3 of imputation. This way, a smaller window size could be applied and the region to scan would be reduced to a very limited percentage of the genome only, resulting in a considerable gain of time.

NOISYmputer is robust on a broad range of samples and its computation time makes it very convenient. Part of the success of NOISYmputer lies in the fact that it performs pre- and post-imputation filtering steps that remove, among other things, incoherent SNPs, meaning SNPs that do not segregate the same way as its immediate environment, often indicating mapping errors. This filtering of incoherent SNPs step uses a Chi-square test to evaluate if the observed pattern is reasonable. Unfortunately, Chi-square test thresholds are dependent on sample sizes. Thus, when imputing many samples (e.g., $m = 2000$) with NOISYmputer, the user has to adapt the Chi-square threshold to the sample size, which is not convenient. A solution to this would be to use a "Cramér's V" statistic instead [13], which would be independent of the sample number in the VCF.

NOISYmputer estimates global error rates for A and B reads for a dataset. However, mapping and sequencing errors are highly variant-specific. So, it might be desirable to estimate A and B reads error rates for each and every variant. This is what GBScleanR, for instance, implements for Genotyping-by-Sequencing data [14]. Nevertheless, in any case, NOISYmputer is already very robust over a wide range of sequencing error rates, while mapping errors are filtered before imputation.

Unlike FSFHap or LB-Impute, NOISYmputer does not impute the parental genotypes, which might result in the loss of SNPs, especially in datasets derived from very low-coverage sequencing. Although we recommend sequencing the parents at high coverage ($\geq$ 20X), it is not always possible—for instance, when re-analyzing historical data. The next version of NOISYmputer will impute the parental genotypes when necessary.

Finally, as pointed in the Results section, the imputation half-window size can have an impact on the outputs of NOISYmputer. NOISYmputer could benefit from an iterative process that would check for different window sizes and analyze the convergence of the results to select the appropriate window size and thus to achieve the best compromise between precision and recall, along with precision of breakpoints positions.

## Supporting information

**S1 File. Options used for imputation with LB-impute, TASSEL-FSFHap and NOISYmputer; Generation of Rice $F_2$ VCF dataset; Cluster description.**
(PDF)

**S2 File. What is the proportion of recombined versus plain diplotypes in an F$_2$ individual?.**
(XLSX)

**S3 File. Calculation details of the loose support interval (PSI).**
(XLSX)

**S1 Fig. Precision and Recall without the parameter values window = 15 and error rate = 0.05.**
(PDF)

**S1 Table. Genetic map sizes of 84 bi-parental populations, varying in marker density, depth and error rate, after imputation with FSFHap and NOISYmputer with default parameters.**
(XLSX)

**S2 Table. Error rate estimation accuracy in 420 simulated VCFs with variable marker density, depth and error rate.**
(XLSX)

**S3 Table. Comparison of estimated with expected map sizes using NOISYmputer, FSFHap and LB-Impute using the VCF filtered by NOISYmputer as input.**
(XLSX)

**S4 Table. Evaluation of NOISYmputer outputs for 20 pseudo 3X samples F$_2$ samples compared to their 20X counterparts.** Pseudo 3X samples are derived from artificially subsetted 20X samples. These pseudo 3X samples were part of the 222 F$_2$ Rice WGS samples imputed using NOISYmputer. Number of SNPs differences are estimated based on the position difference compared to the marker density in the loose support interval.
(XLSX)

**S5 Table. Comparison of numbers of remaining markers after the different filtering steps of NOISYmputer pre-imputation filtering.**
(XLSX)

## Acknowledgments

We thank Karine Labadie (CEA, Institut de Génomique, Genoscope, Evry, France) for sharing the WGS data for the sequencing of rice populations, Christine Tranchant-Dubreuil (IRD, Montpellier, France) for her help with retrieving the Rice_WGS data and François Sabot (IRD, Montpellier, France) for coordinating the IRIGIN project. We are grateful to the Institut Français de Bioinformatique (IFB) for providing computing resources. We also thank the Yale Center for Research Computing for guidance and use of the research computing infrastructure.

## Author Contributions

**Conceptualization:** Cécile Triay, Alice Boizet, Jean-François Rami, Mathias Lorieux.

**Formal analysis:** Cécile Triay, Mathias Lorieux.

**Funding acquisition:** Mathias Lorieux.

**Methodology:** Cécile Triay, Mathias Lorieux.

**Resources:** Jean-François Rami, Mathias Lorieux.

**Software:** Cécile Triay, Alice Boizet, Mathias Lorieux.

**Writing – original draft:** Cécile Triay, Alice Boizet, Mathias Lorieux.

**Writing – review & editing:** Cécile Triay, Alice Boizet, Christopher Fragoso, Anestis Gkano-giannis, Jean-François Rami, Mathias Lorieux.

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
