## [Decision Letter · Decision Letter 0]

23 Jul 2024

PONE-D-24-15414Fast and accurate imputation of genotypes from noisy low-coverage sequencing data in bi-parental populationsPLOS ONE

Dear Dr. Lorieux,

Thank you for submitting your manuscript to PLOS ONE. After careful consideration, we feel that it has merit but does not fully meet PLOS ONE’s publication criteria as it currently stands. Therefore, we invite you to submit a revised version of the manuscript that addresses the points raised during the review process.

The reviewers' comments pointed out that a revision is required to improve the current version of the manuscript. One of the most critical concerns regards the filtering step. It also needs to be clarified if the input of the three tested tools is the same. Please refer to the reviewers’ reports and the Reviewer’s Responses to Questions section for detailed comments, which could help you improve your manuscript. Please carefully address (and reply to) all the comments raised by all reviewers (this is mandatory).

We look forward to receiving your revised manuscript.

Kind regards,

Andrea Tangherloni

Academic Editor

PLOS ONE

Journal Requirements:

"The French ANR project "LANDSREC" 385 (ANR-21-CE20-0012-03), , the French Government France Génomique program through its International RIce 485 Genome INitiative “IRIGIN” project, and the CGIAR Research Program “RICE”"

"We thank Karine Labadie (CEA, Institut de Génomique, Genoscope, Evry, France) for sharing the WGS data for

the sequencing of rice populations, Christine Tranchant-Dubreuil (IRD, Montpellier, France) for her help with

retrieving the Rice_WGS data and François Sabot (IRD, Montpellier, France) for coordinating the IRIGIN

project. We are grateful to the Institut Français de Bioinformatique (IFB) for providing computing resources.

We also thank the Yale Center for Research Computing for guidance and use of the research computing

infrastructure. The following programs supported parts of this initiative: the French ANR project "LANDSREC"

(ANR-21-CE20-0012-03), the French Government France Génomique program through its International RIce

Genome INitiative “IRIGIN” project, and the CGIAR Research Program “RICE”."

"The French ANR project "LANDSREC" 385 (ANR-21-CE20-0012-03), , the French Government France Génomique program through its International RIce 485 Genome INitiative “IRIGIN” project, and the CGIAR Research Program “RICE”"

**Additional Editor Comments:**

The reviewers' comments pointed out that a revision is required to improve the current version of the manuscript. One of the most critical concerns regards the filtering step. It also needs to be clarified if the input of the three tested tools is the same. Please refer to the reviewers’ reports and the Reviewer’s Responses to Questions section for detailed comments, which could help you improve your manuscript. Please carefully address (and reply to) all the comments raised by all reviewers (this is mandatory).

Reviewers' comments:

Reviewer's Responses to Questions

**Comments to the Author**

1. Is the manuscript technically sound, and do the data support the conclusions?

Reviewer #1: Yes

Reviewer #2: Partly

Reviewer #3: Yes

2. Has the statistical analysis been performed appropriately and rigorously? 

Reviewer #1: I Don't Know

Reviewer #2: Yes

Reviewer #3: I Don't Know

3. Have the authors made all data underlying the findings in their manuscript fully available?

Reviewer #1: No

Reviewer #2: Yes

Reviewer #3: Yes

4. Is the manuscript presented in an intelligible fashion and written in standard English?

Reviewer #1: Yes

Reviewer #2: Yes

Reviewer #3: Yes

5. Review Comments to the Author

Reviewer #1: The authors propose a method to genotype bi-parental populations with low-coverage NGS technology and compare their approach against FSFhap and LB-Impute. An MLE algorithm as compared to HMM makes sense with the assumptions of the bi-parental populations with inbred and exclusively homozygous parental lines and the low running time and memory consumption looks good. However, I have some comments/questions as given below:

Major Comments

-------------

- There is a heavy filtering of SNPs before being used by the tool. Filtering multiallelic SNPs and small indels is common in genomic tools (but newer versions of tools are updating their pipelines to incorporate them now) but further filtering of SNPs based on HWE parameters, read counts, etc might be a bit excessive. Do you have any explanation about this? How many SNPs are removed through these filtering criteria (in terms of percentage of all SNPs)?

- Do you compare NOISYmputer to FSFhap and LB-Impute with the input of these filtered variants to all three tools or are the filtering criteria for the three tools different? If variants are filtered differently for the three methods, is it possible to run the three tools on one VCF and compare the results? Is it possible that the high accuracy, better map size estimation, and breakpoint precision of the tool are the outcome of the heavily filtered set of SNPs?

- Line 259: How do you define too many or too few? Is it based on the average read depth or a constant value of X?

Also it is possible that the SNPs with too many reads might be duplicated regions but this information seems to be lost when they are removed. Any comments on this?

- What is the reason that windows in this work are based on the number of variant positions around a central SNP and not base pair distance-based? More clearly, why is an m-window around SNPj defined as SNPj-m to SNPj+m and not all the variants whose base pair distance is less than m from SNPj. Most references have gaps in their sequences and the existing window-system in the manuscript will include variants which can be large distances away from the central SNP if such a gap exists between the two variants.

Why was such a window-system used instead of a base pair distance-based?

- Why was the comparison done with FSFhap and also not with FILLIN (as the main publication of FSFhap does)? FILLIN has imputation capabilities that can also be compared with the imputation done by NOISYmputer.

I could not find a direct link to the simulated and real dataset used in the experiments. Each dataset used in the experiments should be stated clearly with a proper link. For simulations, an open source repository (e.g., Zenodo) might be used.

Minor Comments

------------

- Figure 2 shows that the performance of the algorithm is poor for data with high error rates. So even if the correct error rates are estimated, it does not help in generating accurate results and the tool depends on the underlying error rates.

This is not a very big issue since the tool primarily deals with Illumina reads, which have very low error rates (as stated in lines 354-357). But future ideas related to extending these tools to error-prone long reads will be unsuccessful based on these results. Is it possible to generalize the model further to allow high error rates?

- I would expect to see some stats about the performance of the algorithm in the abstract.

- Line 48 states that “Reducing sequencing costs through minimized per-sample coverage has an important experimental downside: LC-NGS mechanically introduces a series of issues, the main ones being:”

However, I can't see any connection of issue 3 listed in line 60 with the data being low coverage. It's a problem with short-read sequencing.

- Line 117: Where does the equation for the proportion of the genome with no recombination come from? Reference?

- Line 123: “computation time, which increases linearly according to the diplotype size” - Is there any time complexity analysis given as part of the work to show this?

- Line 153-155: Are nAi and nBi numbers or is it a multiplication between n and Ai and Bi. If they are numbers, consider superscipting the A and B. It is not clear how they are being obtained. The binomial expansion has them as variables and their values can range from 0 to ni. But in Line 162 in the equation, they are no longer variables on the LHS of the equation while they exist on the RHS of the equation. Which value is chosen for nAi and nBi and how are they chosen?

- Line 210-225: If BB is the segment to the left of position j and AB is to the right, why is the BB window defined for j to j+m and vice versa for the AB window? Since BB is to the left, the window is for j-m to j. Further definition of PSI as a product of 1-probabilities for the two regions. There is no explanation on why this approach has been taken.

- Line 244-245: The formulation of homozygous to heterozygous transitions seems to be easily generalizable for double recombination events of transitioning from homozygous to homozygous. Why not allow such events as default and allow users to choose whether to consider the double recombination transitions?

- Line 320: Does PoPSimul also simulate gaps in the reference or is it a uniform distribution of SNPs?

- Line 338-342: It can also be considered to report the results using other metrics like precision-recall or Matthews Correlation Co-efficient etc. Maybe even with a confusion matrix.

- Line 316-317: Specs of the cluster are given in the link but which of these machines did you use for the experiments. This information can also be added as a single line regarding the specs of the machine used.

Typos

-----

- Line 102: “leaving unimputed the regions"

- Line 132: “once could set”

- Line 157-159: End bracket missing in the expression in the middle.

- Line 218: Should this be SNPj+m and SNPj-m respectively, where “m” is subscripted?

Reviewer #2: The authors propose a specific scheme for handling imputation of individuals where the original genetic stock come from two completely inbred lines. They propose to compute the total number of reads supporting the two generalized alleles (A and B) and make the primary imputation decisions based on these counts. They claim that this technique will be more tolerant to sequencing and mapping errors compared to other prevalent approaches.

The authors claim that this approach is more efficient than a Hidden Markov Model, but in my mind, what's proposed is similar to a HMM where the emitted symbols are the read counts in each window. Combining windowing with transition probabilities for long range haplotype consistency wouldn't be impossible.

I would also be somewhat weary against the argument that it makes sense to have a single global error rate per origin line, since mapping and sequencing errors are highly variant-specific. On the other hand, I can of course agree that two parameters would maybe be better than a single global epsilon.

The use of the reconstructed chromosome length as a metric for qaulity is ingenious, but it takes some time to wrap your head around it.

Overall, I find the contribution to be interesting, but there are several assumptions made on the type of dataset and I think the manuscript would benefit from more clearly spelling these out and also contrast to related scenarios where they would not be applicable.

Reviewer #3: The paper is written well, I understood the problem of imputation and authors' approach to the maximum likelihood estimation. The results provided are understandable. However, related work or literature review could be improved.

6. PLOS authors have the option to publish the peer review history of their article (what does this mean?). If published, this will include your full peer review and any attached files.

Reviewer #1: No

Reviewer #2: **Yes: **Carl Nettelblad

Reviewer #3: No

---

## [Author Response · Author response to Decision Letter 0]

10 Sep 2024

PLOS ONE – PONE-D-24-15414

Manuscript: “Fast and accurate imputation of genotypes from noisy low-coverage sequencing data in bi-parental populations”

Response to Reviewers

Editor’s comments

> We carefully checked the styles requirements, and hope everything is fine.

> We updated the manuscript using the PLOS LaTeX template.

"The French ANR project "LANDSREC" 385 (ANR-21-CE20-0012-03), the French Government France Génomique program through its International RIce 485 Genome INitiative “IRIGIN” project, and the CGIAR Research Program “RICE”"

> We included the amended role of funders in the cover letter.

"We thank Karine Labadie (CEA, Institut de Génomique, Genoscope, Evry, France) for sharing the WGS data for

the sequencing of rice populations, Christine Tranchant-Dubreuil (IRD, Montpellier, France) for her help with

retrieving the Rice_WGS data and François Sabot (IRD, Montpellier, France) for coordinating the IRIGIN

project. We are grateful to the Institut Français de Bioinformatique (IFB) for providing computing resources.

We also thank the Yale Center for Research Computing for guidance and use of the research computing

infrastructure. The following programs supported parts of this initiative: the French ANR project "LANDSREC"

(ANR-21-CE20-0012-03), the French Government France Génomique program through its International RIce

Genome INitiative “IRIGIN” project, and the CGIAR Research Program “RICE”."

"The French ANR project "LANDSREC" 385 (ANR-21-CE20-0012-03), , the French Government France Génomique program through its International RIce 485 Genome INitiative “IRIGIN” project, and the CGIAR Research Program “RICE”"

> We included the amended statement in the cover letter. Thank you for the help with this.

> All data were made available to download as required.

> This was done.

The reviewers' comments pointed out that a revision is required to improve the current version of the manuscript. One of the most critical concerns regards the filtering step. It also needs to be clarified if the input of the three tested tools is the same. Please refer to the reviewers’ reports and the Reviewer’s Responses to Questions section for detailed comments, which could help you improve your manuscript. Please carefully address (and reply to) all the comments raised by all reviewers (this is mandatory).

> We carefully addressed all the points raised by the Reviewers.

Reviewer #1

Note: all line numbers refer to the file “Revised Manuscript with Track Changes”

The authors propose a method to genotype bi-parental populations with low-coverage NGS technology and compare their approach against FSFhap and LB-Impute. An MLE algorithm as compared to HMM makes sense with the assumptions of the bi-parental populations with inbred and exclusively homozygous parental lines and the low running time and memory consumption looks good. However, I have some comments/questions as given below:

Major Comments

-------------

- There is a heavy filtering of SNPs before being used by the tool. Filtering multiallelic SNPs and small indels is common in genomic tools (but newer versions of tools are updating their pipelines to incorporate them now) but further filtering of SNPs based on HWE parameters, read counts, etc might be a bit excessive. Do you have any explanation about this?

> Thank you for discussing the filtering steps. We do not filter on HWE – since we treat bi-parental populations only – but advanced filtering - before and after imputation - together with MLE-based imputation, is indeed a major feature of NOISYmputer. We filter on genotypic frequencies because some SNPs, generally those which come from erroneous mapping, produce segregations that depart heavily from what is expected in a bi-parental population, even in case of segregation distortion. Keeping those SNPs would add confusion, reducing the difference between the likelihoods of heterozygous and homozygous states. 

The filter on read counts is particularly useful to eliminate SNPs that lay in repeated regions. Keeping them would lead to similar confusion. Furthermore, this is a flexible filter since it is applied per individual, and only the sites whose coverage is at least 10x higher than the average depth (calculated across all sites for the individual) are removed.

The thresholds of these various filters therefore depend on the dataset used and are recalculated to optimally filter the input VCFs. Thus, these “pre-imputation” filters should not be considered as a step preceding NOISYmputer, but as an integral part of the software, contributing to the quality of the imputation results.

Regarding multiallelic sites, they do not correspond to the case of crosses between pure lines, where only two alleles can segregate. This is why they are filtered out. 

- How many SNPs are removed through these filtering criteria (in terms of percentage of all SNPs)?

> We produced a table that we added as Supplemental S5_Table to better inform about the consequences of these filters on the number of remaining sites based on the real datasets analyses and modified main text (lines 423-426) to refer to it.

However, we would like to emphasize that the number of filtered sites heavily depends on the filters that may have been applied during the production of the initial VCF or the sequencing method. Thus, the user might feel that a large number of sites have been lost during NOISYmputer pre-imputation filtering steps if the initial VCF was minimally filtered beforehand and contained noise.

We strongly recommend applying as few arbitrary filters as possible when calling SNPs and producing the initial VCF. For example, in the Snakemake pipeline we provide, we only filter for biallelic sites, sites with at most 80% missing data across the entire population, and sites for which parents are polymorphic and homozygous. These filters are actually applied by NOISYmputer anyway, so applying them beforehand simply allows us to optimize the VCF storage by reducing their final size. No filters on depth, deviation from HWE, or other criteria are applied before running NOISYmputer.

- Do you compare NOISYmputer to FSFhap and LB-Impute with the input of these filtered variants to all three tools or are the filtering criteria for the three tools different? If variants are filtered differently for the three methods, is it possible to run the three tools on one VCF and compare the results? Is it possible that the high accuracy, better map size estimation, and breakpoint precision of the tool are the outcome of the heavily filtered set of SNPs?

> It is correct that NOISYmputer does run filters that FSFhap or LB-Impute do not run, since NOISYMputer’s algorithm is a combination of filtering and imputation. We thus ran again FSFhap and LB-Impute on the real rice and maize datasets pre-filtered by NOISYmputer, and, although the two programs produced better results than without the filtering, they still produced noisy imputed data and therefore extremely long genetic maps. We added Supplementary S3 Table to reflect these new analyses, and modified the text (lines 378-381). We did not re-test FSFhap on simulated data, because the program PopSimul does not produce SNPs with excess counts or incoherent genotype frequencies.

- Line 259: How do you define too many or too few? Is it based on the average read depth or a constant value of X?

Also it is possible that the SNPs with too many reads might be duplicated regions but this information seems to be lost when they are removed. Any comments on this?

> Thank you for the question. It does indeed require some additional explanations. This parameter is not fixed, it varies depending on the dataset used. By default, when reading the input VCF file, if the depth of a site for a sample is 10 times greater than the average depth for that sample (across all its sites), the genotype at that site is set to a missing data. Also by default, no filter is applied on the minimum number of reads required to consider a site in a sample. 

We added an explanation (lines 205-209) in the main text to clarify this section.

This filter is relatively “soft”. For example it corrects only 0.02% of all genotypes as missing data in the Rice F2 dataset (9626/44845342) and 0.002% in the Maize F2 dataset (22/971901).

Regarding the second part of the question, we intentionally filter sites with excessive coverage relative to the sample's average coverage, as it might indicate the presence of (highly) duplicated regions. SNPs from duplicated regions are not informative because we cannot determine which alleles belong to which copy, making it impossible to compare haplotypes to estimate breakpoint positions. 

We have also included this information in the standard output of NOISYmputer and added an output file that now reports the number of sites per sample (datapoints) set as missing data. This feature is currently available on the development branch of our GitLab repository and will be included in the next release of NOISYmputer.

- What is the reason that windows in this work are based on the number of variant positions around a central SNP and not base pair distance-based? More clearly, why is an m-window around SNPj defined as SNPj-m to SNPj+m and not all the variants whose base pair distance is less than m from SNPj. Most references have gaps in their sequences and the existing window-system in the manuscript will include variants which can be large distances away from the central SNP if such a gap exists between the two variants.

Why was such a window-system used instead of a base pair distance-based?

> This is a very interesting question. We chose to use windows based on the number of variant sites - just like FSFhap or LB-Impute do - because the sites are far from being evenly dispersed on the genome. Thus, in base pair-based windows, the number of sites can vary greatly. Since the goal of the windows is to calculate likelihoods, which values depend on the number of sites in the window, using base pair-based windows would produce likelihood values that are difficult to interpret, especially when the number of sites is low. Also, the observed likelihoods depend on the real genotypes, which are modeled by recombination that is only faintly correlated with physical distance. Since the final goal is to calculate accurate genetic maps, in centimorgans, using physical distances do not seem to be appropriate. 

Furthermore, we verified that the site-based method still produces good results when encountering breakpoints in low marker density regions (as shown in the simulated data in main text or in the table below). We looked at the top 25% of largest loose support intervals (which implied that sites are spaced apart) in the pseudo 3X samples of the Rice F2 dataset and observed that the distance to the true position of the breakpoint (in number of SNPs) is the same as in the complete dataset. Finally, even if the physical position in base-pair is higher on average, the median stays very reasonable and the drive of the mean is actually due to a single breakpoint among the 23 analyzed.

Lastly, we report several measures in the breakpoints output file for the user to filter the breakpoint results if needed. For example, we report the size of the loose support interval along with a proxy of a confidence interval, the number of SNPs analyzed and the number of SNPs that have data in the analyzed sample. See the Breakpoints.csv file in NOISYmputer output.

- Why was the comparison done with FSFhap and also not with FILLIN (as the main publication of FSFhap does)? FILLIN has imputation capabilities that can also be compared with the imputation done by NOISYmputer.

> FILLIN, like Beagle or Impute2, was developed to impute genotypes in diversity panels (which the authors call inbred lines libraries). FSFhap, like NOISYmputer, was developed specially to address the problem of bi-parental populations. The authors themselves report that “FSFHap and FILLIN performed very similarly, but FSFHap imputed more heterozygous sites with increased accuracy” in the case of bi-parental populations. We thus considered that FSFhap is well more adapted for comparison with NOISYmputer. 

> I could not find a direct link to the simulated and real dataset used in the experiments. Each dataset used in the experiments should be stated clearly with a proper link. For simulations, an open source repository (e.g., Zenodo) might be used.

> We have made both real and simulated datasets available.

The simulated datasets can be reached at https://zenodo.org/records/13381283, while the real dataset is available at the IRD Dataverse: https://doi.org/10.23708/8FXUNC.

We added this in the main text line ~283 and line ~361.

Minor Comments

------------

- Figure 2 shows that the performance of the algorithm is poor for data with high error rates. So even if the correct error rates are estimated, it does not help in generating accurate results and the tool depends on the underlying error rates. This is not a very big issue since the tool primarily deals with Illumina reads, which have very low error rates (as stated in lines 354-357). But future ideas related to extending these tools to error-prone long reads will be unsuccessful based on these results. Is it possible to generalize the model further to allow high error rates?

> Inde

---

## [Decision Letter · Decision Letter 1]

21 Oct 2024

PONE-D-24-15414R1Fast and accurate imputation of genotypes from noisy low-coverage sequencing data in bi-parental populationsPLOS ONE

Dear Dr. Lorieux,

Thank you for submitting your manuscript to PLOS ONE. After careful consideration, we feel that it has merit but does not fully meet PLOS ONE’s publication criteria as it currently stands. Therefore, we invite you to submit a revised version of the manuscript that addresses the points raised during the review process.

The authors addressed the main Reviewers’ comments; however, a couple of concerns remain that must be considered. For instance, I agree with Reviewer 2 that the authors should improve the repository documentation. Please consider the two minor comments from Reviewer 1.

We look forward to receiving your revised manuscript.

Kind regards,

Andrea Tangherloni

Academic Editor

PLOS ONE

Journal Requirements:

Additional Editor Comments:

The authors addressed the main Reviewers’ comments; however, a couple of concerns remain that must be considered. For instance, I agree with Reviewer 2 that the authors should improve the repository documentation. Please consider the two minor comments from Reviewer 1.

Reviewers' comments:

Reviewer's Responses to Questions

**Comments to the Author**

1. If the authors have adequately addressed your comments raised in a previous round of review and you feel that this manuscript is now acceptable for publication, you may indicate that here to bypass the “Comments to the Author” section, enter your conflict of interest statement in the “Confidential to Editor” section, and submit your "Accept" recommendation.

Reviewer #1: All comments have been addressed

Reviewer #2: All comments have been addressed

2. Is the manuscript technically sound, and do the data support the conclusions?

Reviewer #1: Yes

Reviewer #2: Yes

3. Has the statistical analysis been performed appropriately and rigorously? 

Reviewer #1: Yes

Reviewer #2: Yes

4. Have the authors made all data underlying the findings in their manuscript fully available?

Reviewer #1: Yes

Reviewer #2: Yes

5. Is the manuscript presented in an intelligible fashion and written in standard English?

Reviewer #1: Yes

Reviewer #2: Yes

6. Review Comments to the Author

Reviewer #1: - I think this sentence in the abstract has issues, it should be rephrased: "NOISYmputer is particularly convincing when comparing the genetic map obtained on real datasets as it estimates accurately the size of the map where the two other software can mistake from hundred to hundred thousands centimorgans."

- Link to data in Zenodo should be added under "Availability"

- The code repository will benefit from better documentation about the list of output files that are generated. Multiple output and intermediate files are generated but no explanation for what each file represents have been provided.

Reviewer #2: I still think there would be room for improvement in the presentation, and I can agree with the request from reviewer 3 for additional references. I think even tools such as R/QTL and the original Lander-Botstein model are relevant in the sense that inferring trait status and outright imputation are mirror problems, especially in the biparental case.

7. PLOS authors have the option to publish the peer review history of their article (what does this mean?). If published, this will include your full peer review and any attached files.

Reviewer #1: No

Reviewer #2: **Yes: **Carl Nettelblad

---

## [Author Response · Author response to Decision Letter 1]

23 Oct 2024

Please see the "Response to Reviewers revision 2.pdf" file

---

## [Decision Letter · Decision Letter 2]

18 Nov 2024

Fast and accurate imputation of genotypes from noisy low-coverage sequencing data in bi-parental populations

PONE-D-24-15414R2

Dear Dr. Lorieux,

We’re pleased to inform you that your manuscript has been judged scientifically suitable for publication and will be formally accepted for publication once it meets all outstanding technical requirements.

Kind regards,

Andrea Tangherloni

Academic Editor

PLOS ONE

Additional Editor Comments (optional):

Reviewers' comments:

Reviewer's Responses to Questions

**Comments to the Author**

1. If the authors have adequately addressed your comments raised in a previous round of review and you feel that this manuscript is now acceptable for publication, you may indicate that here to bypass the “Comments to the Author” section, enter your conflict of interest statement in the “Confidential to Editor” section, and submit your "Accept" recommendation.

Reviewer #1: All comments have been addressed

2. Is the manuscript technically sound, and do the data support the conclusions?

Reviewer #1: Yes

3. Has the statistical analysis been performed appropriately and rigorously? 

Reviewer #1: Yes

4. Have the authors made all data underlying the findings in their manuscript fully available?

Reviewer #1: Yes

5. Is the manuscript presented in an intelligible fashion and written in standard English?

Reviewer #1: Yes

6. Review Comments to the Author

Reviewer #1: (No Response)

7. PLOS authors have the option to publish the peer review history of their article (what does this mean?). If published, this will include your full peer review and any attached files.

Reviewer #1: No

---

## [Editor Report · Acceptance letter]

27 Nov 2024

PONE-D-24-15414R2 

PLOS ONE

Dear Dr. Lorieux, 

I'm pleased to inform you that your manuscript has been deemed suitable for publication in PLOS ONE. Congratulations! Your manuscript is now being handed over to our production team.

Kind regards, 

on behalf of

Dr. Andrea Tangherloni 

Academic Editor

PLOS ONE